# A meta-analysis suggests that TMS targeting the hippocampal network selectively improves episodic memory

Elena Badillo Goicoechea[1†], Phillip F Agres[2†], Johanna MH Rau[2], Arantzazu San Agustin[2], Joel L Voss[2*]

[1]Department of Public Health Sciences, The University of Chicago, Chicago, United States; [2]Department of Neurology, The University of Chicago, Chicago, United States

## eLife Assessment

This meta-analysis provides a **fundamental** synthesis of evidence demonstrating that transcranial magnetic stimulation targeting the hippocampal-cortical network reliably enhances episodic memory performance across diverse study designs. The evidence is **convincing**, with rigorous methodology and consistent effects observed despite modest sample sizes and some heterogeneity in stimulation approaches. The work highlights the specificity of memory improvements to hippocampal-dependent memories and identifies key methodological factors-such as individualized targeting-that influence efficacy. Overall, this study offers a timely and integrative framework that will inform both basic memory research and the design of future clinical trials for cognitive enhancement.

*For correspondence:
joelvoss@uchicago.edu

[†]These authors contributed equally to this work

**Competing interest:** The authors declare that no competing interests exist.

**Abstract** Episodic memory is critically dependent on the hippocampal network and is frequently impaired in many clinical disorders. Recent findings highlight Hippocampal Indirectly Targeted Stimulation (HITS) as a promising, network-guided non-invasive transcranial magnetic stimulation (TMS) procedure to enhance episodic memory performance. Here, we report the first comprehensive meta-analysis of HITS effects on episodic memory, encompassing both healthy individuals and clinical populations. HITS using parieto-occipital network targets robustly improved episodic memory, with effects selective for episodic memory versus other non-memory cognitive domains. Efficacy was significantly greater when memory performance was assessed using memory tasks sensitive to recollection, which is strongly linked to hippocampal network function, compared to recognition or other types of episodic memory tasks. Efficacy was also significantly greater when HITS was delivered before the memory tasks were administered versus in the period between study and test phases of tasks. No serious adverse events were reported. These findings establish HITS as a robust approach for episodic memory enhancement, suggesting potential for clinical translation in memory disorders. Selectivity of effects for episodic memory generally and for recollection-format tests in particular indicates cognitive and mechanistic specificity, supporting the potential for targeted and selective neuromodulation of hippocampal networks and their associated functions.

## Introduction

Memory for episodes of experience (episodic memory) depends critically on the hippocampus and interactions of hippocampus with a set of brain regions that comprise a hippocampal-centered network (*Battaglia et al., 2011*; *Braga and Buckner, 2017*; *Dickerson and Eichenbaum, 2010*; *Ferguson et al., 2019*; *King et al., 2015*; *Ritchey et al., 2015*; *Scoville and Milner, 1957*; *Wang et al., 2014*). Episodic memory impairments due to a variety of neurological and psychiatric conditions have been

associated with disrupted functional connectivity of hippocampal-centered networks, as measured via methods, such as functional MRI (*Dickerson and Eichenbaum, 2010*; *Ferguson et al., 2019*; *Ritchey et al., 2015*; *Jalilianhasanpour et al., 2019*; *Tripathi et al., 2025*; *Greicius et al., 2004*; *Garrity et al., 2007*; *Ives-Deliperi and Butler, 2021*; *Joshi et al., 2020*). Thus, to the extent that memory impairments result from disrupted network function, modulation of hippocampal network functional connectivity is a potential therapeutic strategy for episodic memory rescue.

Transcranial magnetic stimulation (TMS) induces electrical fields in the human brain with sufficient intensity to trigger focal action potential firing in neocortex (*Romero et al., 2019*). Repetitive TMS (rTMS) protocols influence neuroplasticity of distributed networks of the stimulated location, which can influence functional connectivity (*Valero-Cabré et al., 2005*; *Fox et al., 2012*; *Chen et al., 2013*; *Eldaief et al., 2011*; *Gratton et al., 2013*; *Liston et al., 2014*). This motivates the hypothesis that rTMS of hippocampal network locations could affect network functional connectivity and thereby influence episodic memory. Indeed, (*Wang et al., 2014*) applied multi-day rTMS to a parietal-cortex location identified via its high fMRI connectivity with the hippocampus. They reported that stimulation increased fMRI connectivity of the indirectly targeted portion of hippocampus with its network and improved episodic memory performance. We refer to this stimulation method as 'Hippocampal Indirectly Targeted Stimulation' (HITS), as the overall goal is to use noninvasive rTMS to indirectly affect hippocampal network interactions. A variety of findings suggest that HITS modulates the activity of brain networks defined relative to the hippocampal indirect target (*Hebscher and Voss, 2020*; *Solomon et al., 2024*). It is possible that HITS could be used to enhance memory function, both in healthy individuals and those with clinical memory impairments resulting from abnormal hippocampal network connectivity (*Jung et al., 2024*; *Nilakantan et al., 2019*; *Tang et al., 2023*). However, the variety of neuromodulation protocols used by various research groups and variability in reported effects of HITS on memory (*Hebscher and Voss, 2020*; *Cash et al., 2022*; *Freedberg et al., 2022b*; *Hermiller et al., 2019*) motivate systematic and quantitative evaluation of this potential.

Here, we report results from a systematic review and meta-analysis of studies that investigated the effects of HITS on episodic memory. We reviewed studies in which rTMS was applied to parieto-occipital locations of the hippocampal network (*Figure 1*) and that assessed effects of stimulation on objective tests of episodic memory, conducted in healthy young and older adults and in clinical populations with memory impairments. Our main analysis goal was to determine whether HITS affects episodic memory reliably across studies, despite variation in study designs. We also tested whether the effects of HITS were greater on episodic memory than on other cognitive functions measured in the same studies, and whether study design factors modulated the effectiveness of HITS. We focused on hypothesis-driven design factors, including task format, targeting method, and study population, as well as eight factors identified *post hoc* as varying across studies (Table in *Supplementary file 1*). Notably, the reviewed studies included a range of design factors, including whether the episodic-memory tasks used recollection versus recognition formats, whether subjects were healthy versus memory impaired, whether the experiments followed acute/basic-science versus chronic/clinical-intervention design models, and other various aspects of trial design. Our analysis goal was to determine whether the effects of HITS on performance were statistically robust irrespective of these factors and whether the experiment designs significantly modulated the effects of HITS.

## Results
### Sample characteristics

Our search identified 47 studies meeting inclusion criteria, 38 of which provided sufficient data to support meta-analysis (*Table 1*). These 38 studies included N=1009 subjects and reported 253 statistical comparisons (i.e. 'effects') of the influence of HITS on performance of episodic-memory (140 effects) and non-memory tasks (113 effects), which we transformed into normalized effects (Hedges' *g*). The mean sample size was n=23.2 for episodic-memory effects (range = 4–68) and n=23.3 for non-memory effects (range = 8–58) (*Figure 1—figure supplement 1*). By definition, stimulation was applied to locations of the hippocampal network that were predominantly in left lateral parietal cortex (84% of studies) or in precuneus or other parieto-occipital neocortex locations generally considered as part of the hippocampal network (all remaining studies) (*Figure 1*).

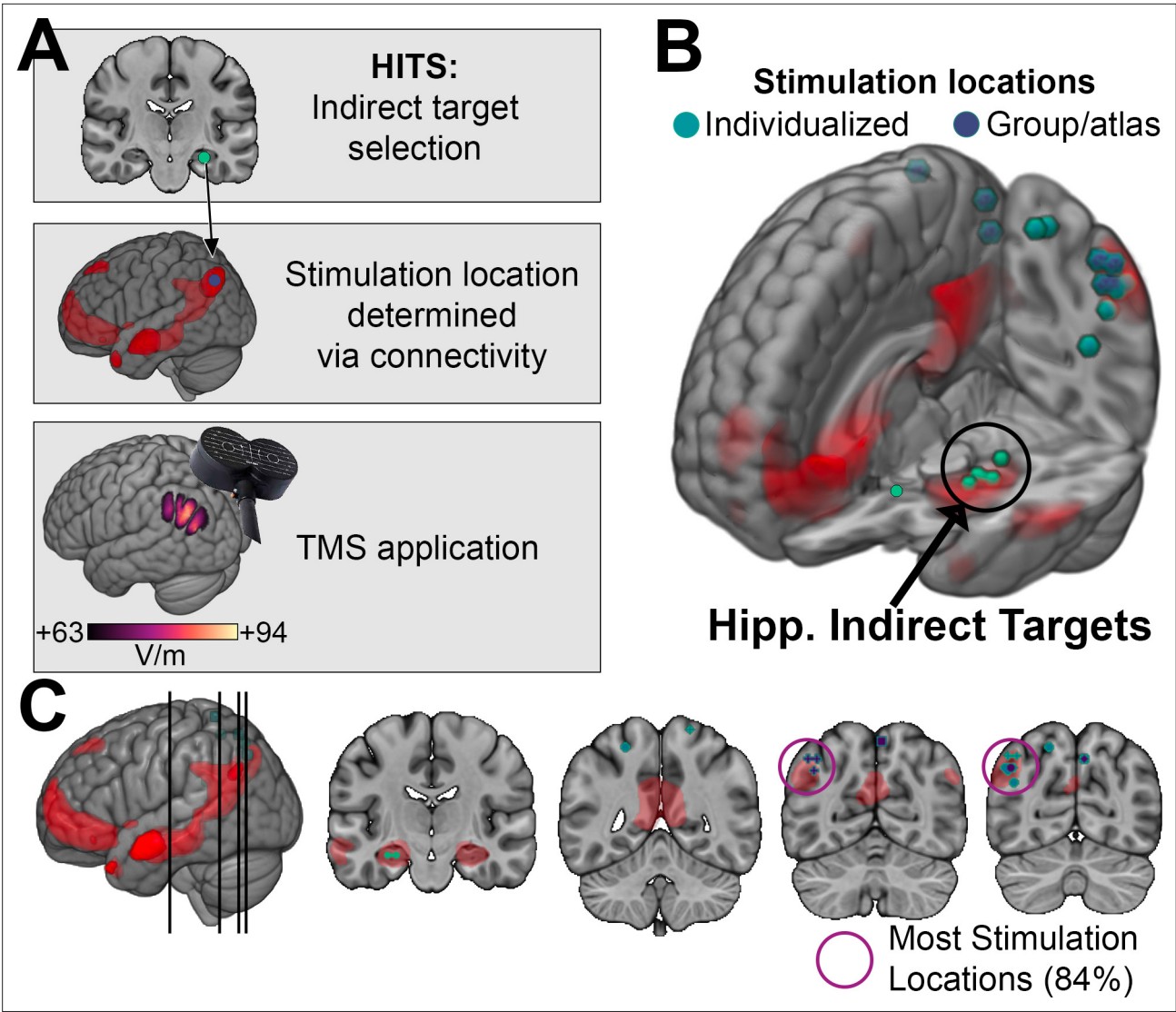

**Figure 1.** Targets and methods used in Hippocampal Indirectly Targeted Stimulation (HITS) experiments. (**A**) The general logic of HITS experiments, whereby hippocampal indirect targets are used to define connectivity networks from which stimulation-accessible locations are selected and stimulated with transcranial magnetic stimulation (TMS). Red coloration indicates an example group-level fMRI connectivity network of the hippocampal indirect target, from which a stimulation location is selected. A representative electrical field (***Saturnino, 2019***) induced by TMS of this location at a typical intensity (estimate of 100% MT) is shown. The field is thresholded at 66% of its maximum intensity, as stimulation of lower intensity should have negligible effects on neuronal activity (all analyzed studies used stimulation intensity of ~70% MT or above). (**B**) Hippocampal indirect targets and neocortical stimulation locations in the analyzed studies are shown overlaid on a template brain. An example resting-state fMRI connectivity network of the hippocampal indirect targets is displayed in red, as in (**A**). Stimulation locations are colorized separately for studies that used individualized targeting versus those that used group/atlas-based targets. For studies using individualized targeting, the average (centroid) location of the targets for all subjects in the study is shown. The hippocampal indirect targets are shown for those studies using individualized targeting. Note that not all stimulation locations fall within the highlighted red hippocampal network, as that specific network is shown for illustrative purposes to highlight proximity of all stimulation targets to a typical group-defined hippocampal network. (**C**) Coronal slices for the indicated positions show the hippocampal indirect targets and stimulation locations in greater detail. K-means clustering indicated that the majority of stimulation locations (84%) comprised a cluster within left parietal cortex, as indicated.

The online version of this article includes the following figure supplement(s) for figure 1:

**Figure supplement 1.** Histogram of sample sizes per analyzed effect, plotted separately for episodic-memory and non-memory effects.

**Table 1.** Studies identified by systematic review.

Study information is listed separately for studies that were included versus excluded from meta-analyses. Sample sizes listed are for the number of subjects contributing to statistical effects that were analyzed or described.

| Study # | First author | Year | Sample Size | DOI |
|---|---|---|---|---|
| *Included in meta-analysis* | | | | |
| 1 | Wang | 2014 | 16 | 10.1126/science.1252900 |
| 2 | Yazar | 2014 | 69 | 10.1371/journal.pone.0110414 |
| 3 | Wang | 2015 | * | 10.1002/hipo.22416 |
| 4 | Bonnì | 2015 | 30 | 10.1016/j.bbr.2014.12.032 |
| 5 | Nilakantan | 2017 | 16 | 10.1016/j.cub.2016.12.042 |
| 6 | Yazar | 2017 | 23 | 10.1016/j.brs.2017.02.011 |
| 7 | Kim | 2018 | 32 | 10.1126/sciadv.aar2768 |
| 8 | Tambini | 2018 | 22 | 10.1162/jocn_a_01300 |
| 9 | Bonnici | 2018 | 22 | 10.1523/JNEUROSCI.1239-18.2018 |
| 10 | Koch | 2018 | 14 | 10.1016/j.neuroimage.2017.12.048 |
| 11 | Wynn | 2018 | 19 | 10.1101/lm.048033.118 |
| 12 | Ye | 2018 | 18 | 10.1523/JNEUROSCI.0660-18.2018 |
| 13 | Hermiller | 2019 | 24 | 10.1002/hipo.23054 |
| 14 | Hermiller | 2019 | 14 | 10.1002/brb3.1393 |
| 15 | Nilakantan | 2019 | 15 | 10.1212/WNL.0000000000007502 |
| 16 | Hermiller | 2020 | 16 | 10.1523/JNEUROSCI.0486-20.2020 |
| 17 | Chen | 2020 | 13 | 10.18632/aging.202313 |
| 18 | Gao | 2021 | 32 | 10.1016/j.brainres.2021.147510 |
| 19 | Hebscher | 2021 | 20 | 10.1016/j.cub.2021.01.027 |
| 20 | Freedberg | 2021 | 23 | 10.1016/j.neuroimage.2021.118199 |
| 21 | Velioğlu | 2021 | 15 | 10.1016/j.nlm.2021.107410 |
| 22 | Jia | 2021 | 69 | 10.3389/fnagi.2021.693611 |
| 23 | Freedberg | 2022 | 29 | 10.1016/j.bbr.2021.113707 |
| 24 | Dave | 2022 | 16 | 10.1016/j.crneur.2022.100030 |
| 25 | Hua | 2022 | 39 | 10.3389/fnhum.2022.973298 |
| 26 | Hermiller | 2022 | 30 | 10.1016/j.neurobiolaging.2021.09.018 |
| 27 | Chen | 2022 | 8 | 10.1073/pnas.2113778119 |
| 28 | Wei | 2022 | 29 | 10.1016/j.psychres.2022.114721 |
| 29 | Tang | 2023 | 89 | 10.1093/schbul/sbad015 |
| 30 | Chen | 2023 | 18 | 10.1111/cns.14177 |
| 31 | You | 2023 | 34 | 10.2147/CIA.S416992 |
| 32 | Jin | 2024 | 11 | 10.1016/j.bspc.2023.105725 |
| 33 | Cheng | 2025 | 15 | 10.1162/jocn_a_02273 |
| 34 | Jung | 2024 | 30 | 10.1001/jamanetworkopen.2024.9220 |
| 35 | Lv | 2024 | 60 | 10.1016/j.bbr.2024.115117 |
| 36 | Webler | 2024 | 24 | 10.1016/j.bpsgos.2024.100309 |

*Table 1 continued on next page*

*Table 1 continued*

| Study # | First author | Year | Sample Size | DOI |
|---|---|---|---|---|
| 37 | Velioğlu | 2024 | 12 | 10.29399/npa.28420 |
| 38 | Zheng | 2024 | 43 | 10.1093/cercor/bhae460 |
| *Excluded from meta-analysis* | | | | |
| 39 | Zhao | 2016 | 30 | 10.18632/oncotarget.13060 |
| 40 | Warren | 2019 | * | 10.7554/eLife.49458 |
| 41 | Hendrikse | 2020 | 39 | 10.1016/j.cortex.2020.08.028 |
| 42 | Wang | 2020 | 8 | 10.3389/fnhum.2020.541791 |
| 43 | Cash | 2022 | * | 10.1016/j.brs.2022.09.004 |
| 44 | Yang | 2022 | 16 | 10.3233/JAD-215390 |
| 45 | Chen | 2022 | 12 | 10.3233/JAD-210661 |
| 46 | Li | 2023 | 28 | 10.3390/brainsci13030419 |
| 47 | Li | 2024 | 61 | 10.2147/NDT.S468219 |
| 48 | Koen | 2018 | 20 | 10.1080/17588928.2018.1484723 |

*Studies that re-analyzed data from another study and, therefore, did not contribute independent data are indicated.

## HITS improved episodic memory performance

A meta-regression model with a random effect by study was used to evaluate the overall effect of HITS on episodic memory performance for the 140 effects derived from episodic-memory tasks. The overall effect was positive and highly significant, indicating that HITS improved episodic memory (Hedges' $g=0.44$; 95% CI [0.34, 0.54]; $p<0.001$; *Figure 2A*). A similar result was obtained when outliers were removed in a sensitivity analysis (Hedges' $g=0.38$; 95% CI [0.29, 0.46]; $p<0.001$). Both the main analysis and sensitivity analysis showed lower bounds well above conventional thresholds for small effects ($g \geq 0.2$), thus indicating robust evidence (*Sullivan and Feinn, 2012*) for meaningful improvement in episodic memory performance due to HITS.

## Effects of HITS were greater on episodic memory than non-memory tasks

Meta-regression was used to separately evaluate the overall effect of HITS on performance for 113 effects that concerned tasks assessing cognitive functions other than episodic memory. These included tasks designed to assess attention, working memory, executive functions, and language. The overall effect was almost zero and not significant, indicating that HITS did not reliably affect performance in these tasks (Hedges' $g=0.04$, 95% CI [–0.01, 0.09], $p=0.12$; *Figure 2B*). A similar result was obtained when outliers were removed in a sensitivity analysis ($g=0.01$, 95% CI [–0.05 0.06]; $p=0.79$).

To more rigorously assess whether the effects of HITS for tasks that measured episodic memory were significantly greater than for non-memory tasks, we ran a model pooling all 253 memory and non-memory effects and assessing overall moderation by task type. Effects for non-memory tasks were significantly less than for episodic-memory tasks (Hedges' $g$ modification = –0.39; 95% CI [-0.48,–0.29]; $p<0.001$). A similar result was obtained when outliers were removed in a sensitivity analysis (Hedges' $g$ modification = –0.34; 95% CI [-0.43–0.25]; $p<0.001$). Thus, the beneficial effects of HITS were significantly greater for tasks measuring episodic memory than other cognitive domains in both the main and sensitivity analyses, with no evidence that HITS affected tasks measuring other cognitive domains.

## Study factors modulated HITS effects on episodic memory

We categorized episodic-memory effects according to study design factors, based on a priori hypotheses as well as *post hoc* considerations of variability in study designs (Table in *Supplementary file 1*). We evaluated which of these factors had the potential for independent modulation of HITS effects on

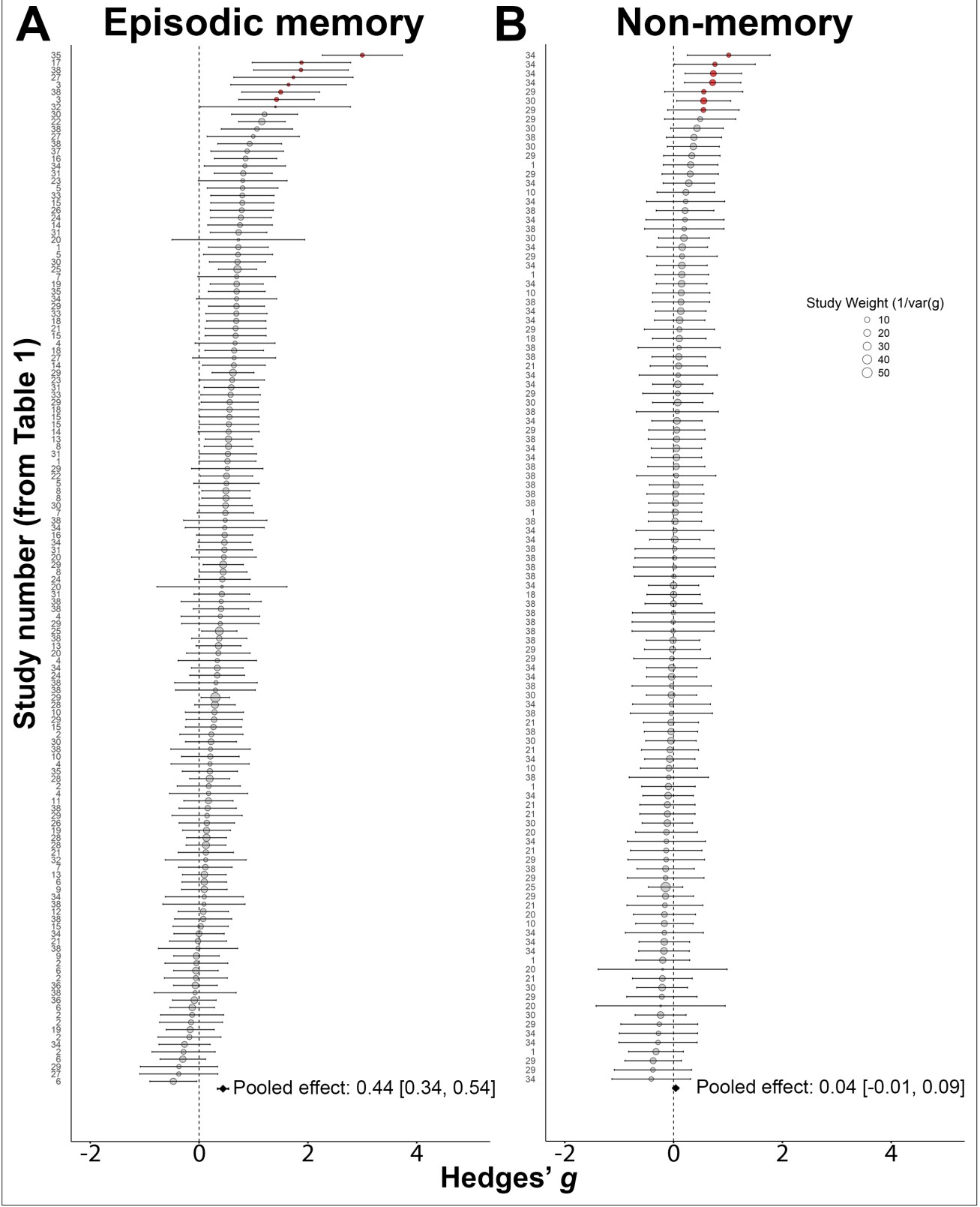

**Figure 2.** Hippocampal Indirectly Targeted Stimulation (HITS) selectively improved episodic memory overall. (**A**) Forest plot of all effects of HITS on episodic-memory task outcomes, ordered by size. (**B**) Forest plot of all effects of HITS on non-memory task outcomes. The pooled effect is shown for both plots. Red circles indicate outliers that were excluded in sensitivity analyses. Circle size indicates the study's weighted contribution to the meta-regression model.

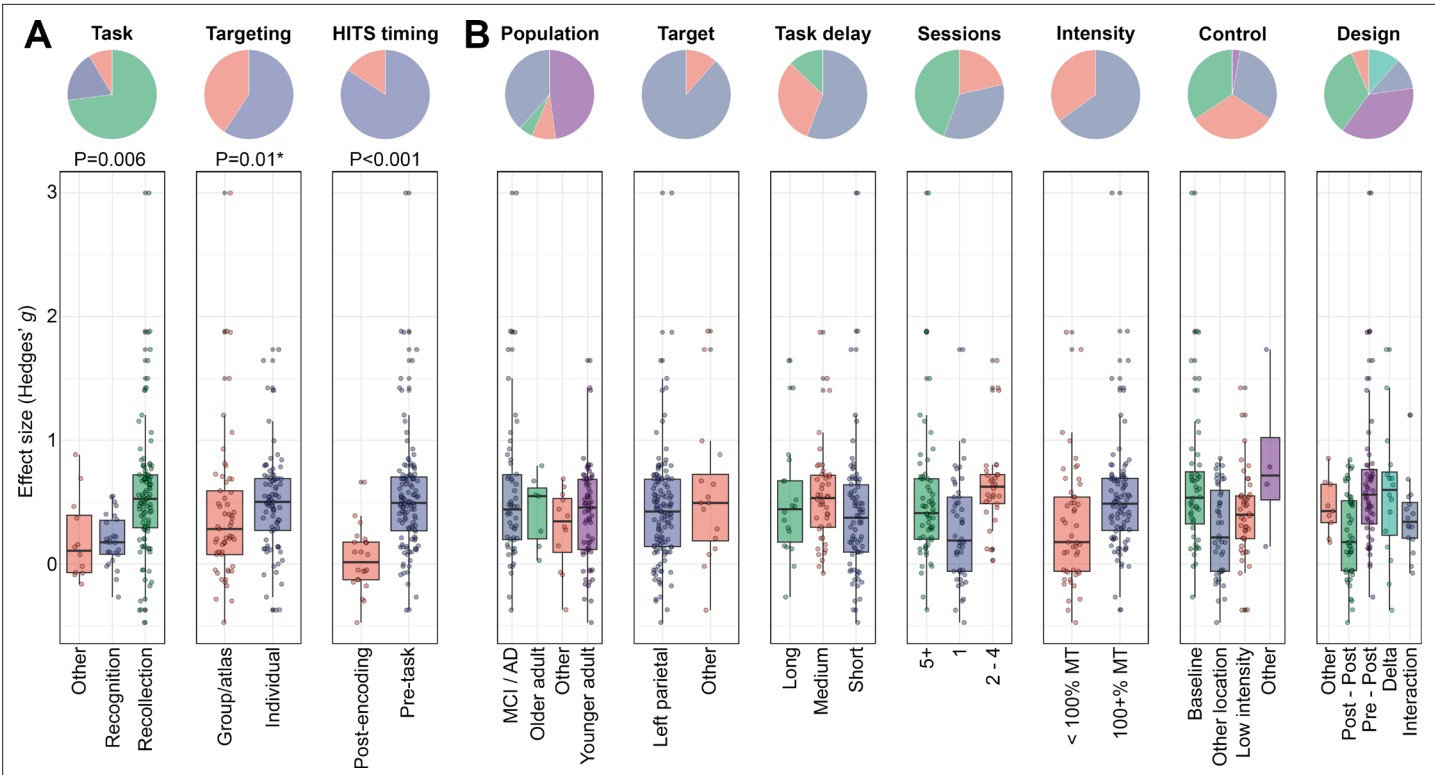

**Figure 3.** Study factors that modulated effects of Hippocampal Indirectly Targeted Stimulation (HITS) on episodic memory. (**A**) Factors that significantly modulated HITS effects on episodic memory. The pie chart indicates the percentage of effects for each category of each factor. Individual effects are shown in the box plots, grouped by factor levels. (**B**) Factors that did not significantly modulate HITS effects on episodic memory, plotted in the same format. *Indicates an effect modification that was significant in the main analysis but not in the sensitivity analysis.

The online version of this article includes the following figure supplement(s) for figure 3:

**Figure supplement 1.** Relationships among factors in episodic memory studies.

**Figure supplement 2.** Funnel plot of episodic memory Hippocampal Indirectly Targeted Stimulation (HITS) effects.

**Figure supplement 3.** Funnel plot of non-memory Hippocampal Indirectly Targeted Stimulation (HITS) effects.

episodic memory via cross-validated lasso regression selection with permutation. Factors that were retained in greater than 50% of the lasso iterations were included in the meta-regression. As expected, the 10 retained factors were well represented among analyzed effects (*Figure 3*). The TMS Protocol factor was excluded, likely because there was strong redundancy of this factor with the Intensity and Sessions factors (*Figure 3—figure supplement 1*). In the sensitivity analysis with outlier removal, TMS Protocol was also discarded, as well as the Task Delay factor. Three of the retained factors significantly modulated the effects of HITS on episodic memory performance (*Figure 3A*), as described next.

Consistent with our a priori hypothesis, effects for tasks that measured episodic memory using recollection test formats were significantly greater than for those that used recognition formats (Hedges' *g* modification = 0.23 greater for recollection versus recognition; 95% CI [0.08, 0.39]; *p*=0.004; *Figure 3A*). In contrast, effects for tests that used other episodic memory (non-recollection and non-recognition) formats were not significantly different from recognition (Hedges' g modification = 0.07 less for other-format versus recognition; 95% CI [–0.33, 0.20]; *p*=0.62; *Figure 3A*). The same pattern held in the sensitivity analysis with outliers removed (Hedges' g modification = 0.20 greater for recollection than recognition; 95% CI [0.07, 0.34]; *p*=0.003; Hedges' g=0.01 less for other-format than recognition; 95% CI [–.24, 0.23]; *p*=0.96). Thus, among tasks assessing episodic memory, the effects of HITS were significantly greater for recollection.

Regarding the Timing factor, HITS applied before the task (from days to seconds before) was associated with significantly greater effects on episodic memory than HITS applied immediately after the period of memory encoding and before the retrieval test (Hedges' g modification = 0.68 greater for pre-task versus post-encoding; 95% CI [0.34, 1.02]; *p*<0.001; *Figure 3A*). The same pattern held

in a sensitivity analysis with outliers removed (Hedges' g modification = 0.59 greater for pre-task versus post-encoding; 95% CI [0.32, 0.87]; *p*<0.001). The average effect for studies in which HITS was applied post-encoding was nearly zero (***Figure 3A***), with these being the majority (52.4%) of all observed negative effects (i.e. effects reflecting impairment by HITS). In contrast, effects when HITS was applied pre-task were almost all positive (91.5% of pre-encoding effects were positive). Notably, studies in the post-encoding category all followed the same general experiment design, whereby a brief encoding period was immediately followed by a single session of stimulation which was then immediately followed by the test. In contrast, pre-task studies included a variety of designs, including those in which brief stimulation trains were given immediately before individual memoranda as well as those in which many days or weeks of stimulation were given for days to weeks before memory encoding (Table in ***Supplementary file 1***). Notably, this variability in the number of sessions was explicitly assessed via the Sessions factor (Table in ***Supplementary file 1***). Thus, among tasks assessing episodic memory, the effects of HITS were significantly greater when it was applied sometime before the task, and thus before the period of memory encoding.

Regarding the Targeting factor, effects differed for studies that used individualized (i.e. subject-specific) measures of functional or structural MRI connectivity to determine the stimulation location for HITS compared to studies that used a similar target for all subjects based on an atlas or a priori hypothesis (Hedges' g modification = –0.30; 95% CI [–0.54,–0.06]; *p*=0.013; ***Figure 3A***). However, this effect was not significant in a sensitivity analysis with outliers removed (Hedges' g modification = –0.16; 95% CI [–0.36, 0.03]; *p*=0.09). Thus, evidence for modification of HITS effects by targeting method was weak, not surviving sensitivity analysis.

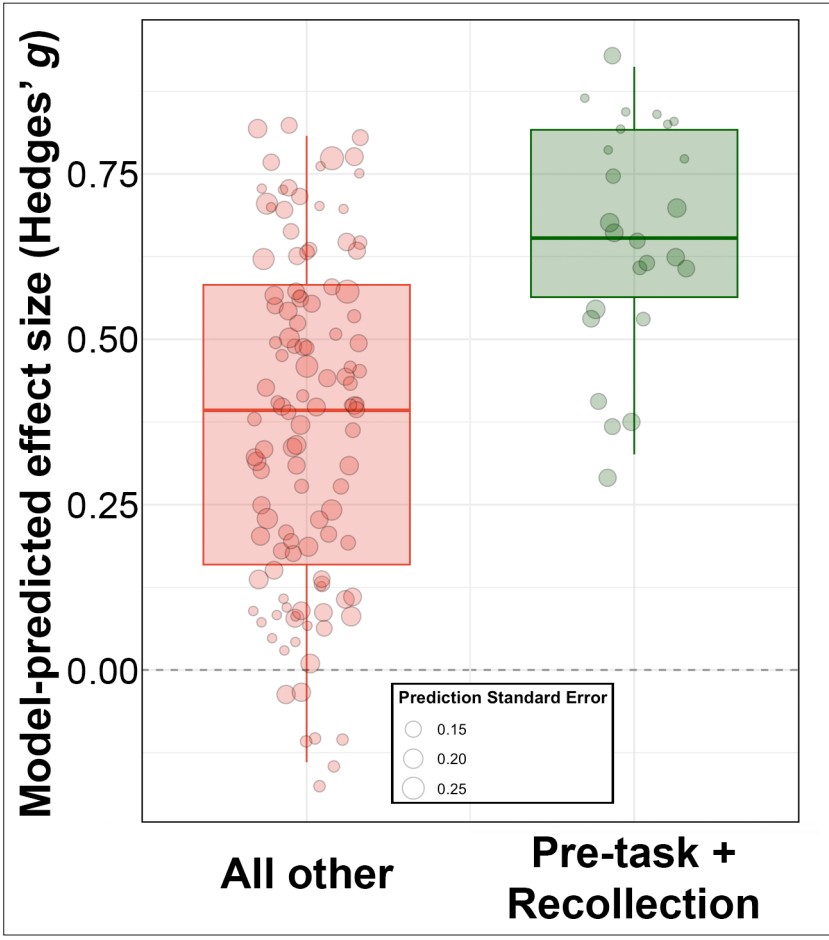

**Figure 4.** Greater effects of Hippocampal Indirectly Targeted Stimulation (HITS) on memory in studies having optimized designs. Box plots of expected effect sizes projected from the meta-regression model for studies in which HITS was applied pre-task and measured using recollection format tests, versus all other study factors. Circles are individual projected effects, with circle size indicating standard error of the prediction, as indicated.

None of the other study factors significantly modulated the effects of HITS in either the main or sensitivity analyses (*Figure 3B*).

Of the 140 effects concerning episodic memory, 17.9% involved both the two significant positive moderating study factors identified in the main and sensitivity analyses (recollection test format and application of HITS before encoding). These effects thus can be considered as reflecting optimized experiment designs. To estimate the expected effects for future studies incorporating both of these factors, we used our meta-regression model to project the expected effect sizes for studies incorporating these factors while accounting for variance associated with other factors held constant. Their mean effect estimate was g=0.66 (95% CI [0.51, 0.82]), in contrast to that of all other effects incorporating non-optimized factors of all other effects, g=0.36 (95% CI [0.29, 0.44]) (*Figure 4*).

## No serious adverse events were reported

None of the reviewed studies reported any unexpected or serious adverse events. The total sample reviewed included N=1223 subjects, including healthy younger and older adults, individuals with mild to moderate Alzheimer's dementia, and individuals with psychiatric symptoms. This suggests that the incidence of serious adverse events due to HITS is low, which is consistent with the general evidence for very low incidence of serious adverse events due to rTMS (*Lerner et al., 2019*).

## Similar outcomes in studies that were excluded from meta-analysis

Ten studies were excluded from meta-analysis (*Table 1*). Excluded studies reported outcomes from 214 subjects and incorporated study design factors representative of those for included studies. Five of these exclusions were because statistical information provided in the publications were not sufficient for calculation of standardized effect sizes, and authors could not be reached to provide additional information (studies 39, 42, 44, 45, 47 from *Table 1*). One exclusion (study 40) was made because the sample of subjects analyzed overlapped 94% with subjects that contributed data to a previous publication that was included, and so exclusion was necessary to avoid redundant statistical results in the meta-analysis. Two studies were excluded because they did not report the effects of HITS on the performance of an episodic memory test, but instead quantified the correlation between fMRI connectivity and the effects of HITS on episodic memory performance (studies 43 and 46). One study was excluded (study 41) because re-analysis of the same data by the same research team (study 43) reported a different conclusion than the original report, suggesting that stimulation may not have been delivered to intended hippocampal network targets in the original report, and thereby calling into question findings from the original report and making it unclear how to categorize the Targeting or Target factors for meta-analysis. One study was excluded (study 48) because HITS was applied 'online,' simultaneous with the trials in which memoranda were shown during the encoding period, whereas all other reviewed HITS studies used 'offline' stimulation, with HITS applied sometime before or after the task period, and therefore it would have been problematic to categorize this study for the Timing factor.

Of the nine excluded studies, eight reported the effects of HITS on episodic memory performance (studies 39, 40, 41, 42, 44, 45, 47, 48). Five of these eight reported that HITS significantly improved performance in at least one of the episodic memory tasks that were administered (studies 39, 40, 42, 44, 45). One of those studies (study 44) also reported numeric but non-significant decrease in memory performance in a subgroup analysis of subjects with mild Alzheimer's dementia. Two of the eight studies (41 and 47) reported small numeric improvements in memory following HITS that were not statistically significant. One of the eight studies (*Freedberg et al., 2022a*) reported memory performance following HITS that was almost identical to that in the control-stimulation (vertex) condition. Notably, study 48 used 'online' stimulation during memory encoding, which has generally been associated with performance impairment (e.g. References *Yeh and Rose, 2019*; *Beynel et al., 2019*).

Regarding the study factors that modulated the effects of HITS in the main study (*Figure 3A*), the eight studies all included at least one test using a recollection format (Task factor: Recollection) and nine administered HITS prior to the memory task (Timing factor: pre-task). Two used individualized targeting (studies 40 and 41), with the others using group/atlas-based targeting. Other factors that were not significantly related to memory outcomes following HITS in the main analysis (*Figure 3B*) varied among these seven excluded studies.

The other two excluded studies (43 and 46) investigated correlations between resting-state fMRI connectivity and the effects of HITS on episodic memory, without directly assessing the impact of HITS on episodic memory performance. One study reported that fMRI connectivity of the hippocampus to the stimulated location predicted episodic memory improvement due to HITS (study 43). The other study reported that HITS changed the correlation between fMRI connectivity and memory performance scores (study 46).

Overall, reported results in the excluded studies were generally consistent with findings from the meta-analysis of included studies. That is, of the eight excluded studies that reported effects of HITS on episodic memory, the majority (5/8) reported statistically significant episodic memory enhancement by HITS, but with variability in effect sizes among studies and a minority (2/8) reporting small but statistically nonsignificant improvement in memory due to HITS. Thus, if effects from these studies could have been included in the meta-analysis, it is unlikely that conclusions would have been substantially affected.

## Discussion

Enhancement of episodic memory in humans is an important scientific objective with clear implications for the treatment of clinical memory disorders and for enhancing function in healthy individuals, including older adults experiencing normative memory decline. However, interventions for memory enhancement have been elusive. Our findings indicate HITS is a promising method to achieve memory enhancement via noninvasive stimulation. There were similarly positive effects among participant populations, from young healthy adults to those with memory disorders, and for a wide range of assessment delays, from seconds to weeks after subjects received HITS. In contrast, there was essentially no effect on non-memory functions, suggesting that memory enhancement did not come at the cost of any off-target cognitive detriment. There were no reported unexpected or serious adverse events. Findings suggest that HITS can be used to achieve robust and selective enhancement of episodic memory.

Many have speculated that brain stimulation targeting a given network can have specific effects on the cognitive abilities hypothesized to depend on that network (*Fox et al., 2012*; *Hebscher and Voss, 2020*; *Cash et al., 2022*; *Pascual-Leone et al., 2000*; *Polanía et al., 2018*). Although aspects of this hypothesis for HITS have been supported in isolated experiments (*Wang et al., 2014*; *Warren et al., 2019*; *Kim et al., 2018*), the current findings are notable in providing meta-analytic support. The hippocampus and its network are strongly implicated in episodic memory, and particularly in tests that measure recollection compared to recognition (*Eichenbaum et al., 2007*; *Yonelinas, 2002*; *Barnett et al., 2021*). Indeed, we found that effects of HITS were specific to episodic memory versus other cognitive tests and were greater for recollection than recognition or other-format memory tests. Although previous individual experiments have shown that HITS affects activity of the hippocampal network in relation to improved recollection more so than recognition (*Nilakantan et al., 2019*; *Kim et al., 2018*; *Nilakantan et al., 2017*), the current findings provide strong evidence that across many different experiment parameters, stimulation targeting the hippocampal network specifically affects the memory function thought to depend most heavily on this network.

Specificity of HITS effects on recollection is mechanistically informative. For instance, other interventions, such as transcranial electrical stimulation yield nonspecific effects across a variety of cognitive domains, including when applied to the lateral parietal areas overlapping with those reviewed here (*Grover et al., 2023*). It is possible that such nonspecific effects could arise via modulation of general non-targeted factors, such as arousal via effects of stimulation on either peripheral or central nervous systems (*van Boekholdt et al., 2021*; *Majdi et al., 2023*). Such nonspecific influences could be problematic, as general and/or off-target effects could be ineffective or detrimental when applied to disorders affecting specific brain circuits. Furthermore, TMS of the frontal cortex developed for anti-depressant intervention has been reported to affect symptoms of a variety of psychiatric disorders and to nonspecifically affect performance of tasks measuring a variety of cognitive abilities (*Kan et al., 2023*). Along with previous neuroimaging evidence for selectivity (*Wang et al., 2014*; *Nilakantan et al., 2019*; *Warren et al., 2019*; *Kim et al., 2018*; *Freedberg et al., 2019*; *Hermiller et al., 2020*), the current findings suggest that HITS is suitable for targeted interventions of hippocampal network contributions to recollection. This could be useful in the treatment of memory disorders, which tend to disproportionately impact recollection relative to recognition (*Yonelinas,*

*2002*). However, an important caveat is that the tests used as outcomes in the studies we analyzed were primarily designed to measure different cognitive abilities in isolation in the context of neuro-psychological diagnosis. Our findings thus do not preclude off-target effects in specialized tasks that deliberately assess the interactions of brain networks. That is, enhancement of episodic memory could affect tests of other cognitive domains when those tests are at least partially sensitive to, or compete with, episodic memory (*Freedberg et al., 2022a*). Nonetheless, the current results support HITS as notable among existing neuromodulatory approaches for producing selective effects on the intended cognitive function, with robust evidence for reproduction of effects supported via meta-analysis.

It is notable that HITS had little to no effect in experiments that delivered a single session of HITS immediately after an encoding session and before the corresponding memory test (*Figure 3A*, HITS Timing factor of Post-Encoding). In contrast, effects were robustly positive when HITS was applied at some point before the task, and thus before memory encoding, across a variety of potential stimulation-encoding delay intervals. This suggests that HITS improves memory formation rather than retention of recently learned information or memory retrieval. This is consistent with findings that hippocampus makes distinct contributions to learning versus retrieval (*Gedankien et al., 2025*). However, an important caveat to this interpretation is that studies did not attempt to cleanly disso-ciate effects of HITS on encoding versus retrieval, and so any interpretation of selectivity of HITS effects on encoding should be made with caution. Regarding HITS as a potential intervention, it is notable that effects were similar across a variety of delays between when HITS was applied and when memory encoding occurred (*Figure 3B*, Task Delay factor). Unlike potential interventions that require delivery of stimulation concurrent with cognitive performance (e.g. transcranial electrical stimulation) or with highly specific information about the timing or successfulness of specific memory events (e.g. closed-loop brain stimulation), this finding suggests that HITS does not require specific knowledge of when memory demands occur following intervention. This is of practical utility, as it is difficult to know a priori when memory encoding will be required during ongoing activities of daily living.

It is important to avoid over-interpretation of null findings concerning moderating effects of some study factors (*Figure 3B*), as many were correlated (*Figure 3—figure supplement 1*) and were selected in individual experiments based on a priori hypotheses. For instance, based on evidence that stimu-lation for more consecutive sessions of HITS leads to greater or more persistent effects (*Freedberg et al., 2020*), studies that included more HITS sessions also tended to measure outcomes after longer delays (correlation of Sessions and Task Delay factors; *Figure 3—figure supplement 1*). Thus, it is possible that studies with more stimulation sessions would have yielded greater effects had they used shorter testing delays, as in the studies with fewer stimulation sessions. Likewise, the Intensity factor was complicated in that studies using lower intensity were more likely to involve theta-burst stimula-tion than studies using higher intensity, which typically used high-frequency rTMS. These studies were based on hypotheses that hippocampal networks should respond to theta-burst stimulation more than high-frequency stimulation (*Hebscher and Voss, 2020*; *Hermiller et al., 2020*). However, theta-burst stimulation typically used lower stimulation intensity and also fewer pulses of stimulation than high-frequency rTMS. Thus, no differences in outcome based on the Intensity factor could support the hypothesis that theta-burst stimulation is more effective than rTMS, as similar effect sizes resulted with fewer and less intense stimulation pulses. Interpretation of null effect modulation by factors that are highly correlated with other factors is thus ambiguous. Although it was useful to identify those factors that robustly modulated effects of HITS via meta-analysis (*Figure 4*), future experimental studies will be needed to systematically assess whether effects of HITS vary by study design (*Figure 3B*), based on specific manipulation of individual factors while holding other factors constant.

A limitation of our approach is that we focused on studies applying HITS to lateral and medial parieto-occipital neocortex targets of the hippocampal network. Other prominent locations of this network include portions of medial frontal and lateral temporal neocortex (*Kahn et al., 2008*). However, no studies targeting these locations were identified by our review, potentially because elec-trical fields generated by conventional TMS are not sufficiently intense or focal or because stimulation is not well tolerated for these targets. Notably, the hippocampal network includes portions of lateral cerebellar cortex (*Buckner et al., 2011*), and one study identified by our review found episodic memory improvement following theta-burst TMS of this location (*Dave et al., 2020*). Theta-burst stim-ulation of lateral cerebellum has also been shown to affect hippocampal network fMRI connectivity (*Halko et al., 2014*). However, this study was excluded from meta-analysis to avoid heterogeneity in

stimulation locations. Future work could systematically evaluate alternative stimulation targets within the hippocampal network, including those that may require stimulation methods other than TMS.

The current findings suggest that HITS has robust effects on episodic memory and that studies, including optimized parameters yield especially robust effects (*Figure 4*). However, further research would be needed to test clinical efficacy for any specific conditions and to determine mechanisms of action. Although our review included studies of patients with episodic memory impairments due to mild to moderate dementia and psychiatric symptoms, the studies did not assess whether gains by HITS could be maintained for months to years or whether they were clinically meaningful. It is also not clear whether HITS will be useful for memory impairments in disorders other than those included in our review, which may involve distinct mechanisms, or whether there is a threshold for degradation of functional and/or structural brain networks beyond which HITS would no longer be effective. We consider our findings as motivating for future attempts to evaluate HITS safety and efficacy for specific clinical indications. Implementations for specific disorders will likely require better understanding of how HITS affects neural function, such as could be obtained by directly recording hippocampal activity while HITS is applied (*Hermiller et al., 2020*; *Li et al., 2025*), in order to tailor HITS to target disorder-specific neural impairments. The current findings encourage such future explorations.

## Materials and methods

### Eligibility criteria

We included reports published between September 2014 (date of the seminal HITS publication *Wang et al., 2014*) and April 2025 that: (1) used rTMS to stimulate hippocampal network locations and (2) included at least one objectively scored test of episodic memory as an outcome. Studies were considered as stimulating hippocampal network locations if they delivered rTMS to: (1) an individual-ized/subject-specific location defined by fMRI connectivity analysis (e.g. a neocortical location iden-tified by resting-state fMRI connectivity to hippocampus in an individual), (2) a location based on a group metric of hippocampal fMRI connectivity (e.g. a neocortical location identified in a group-based atlas of hippocampal resting-state fMRI connectivity), and (3) a similar location based on a hypothesis (e.g. a location that is in a group atlas of hippocampal connectivity but that was selected to test a hypothesis about its function). We limited spatial variability among studies by including only those that stimulated parietal and superior occipital neocortex. We excluded studies using other potential hippocampal network locations due to their rarity, including one study with cerebellar cortex stimula-tion (*Dave et al., 2020*). Studies were considered as including an objective test of episodic memory if at least one of the reported outcomes used a memory test that allowed unambiguous computation of the effect of stimulation on performance accuracy (i.e. experimenter-controlled stimulus set presented to subjects for encoding, for which memory was subsequently tested after some retention interval). When studies also included tests of episodic memory that could not be unambiguously scored (i.e. subjective reports that could not be verified for accuracy), only the objectively scorable tests were included in our analysis.

### Search strategy

Two publicly available online scientific databases were searched: PubMed (pubmed.ncbi.nlm.nih. gov) and Scopus (scopus.com). These databases were most recently searched on April 26, 2025. The primary search was conducted with PubMed, using the PubMed Advanced Search Builder. Two search terms of interest were entered: 'memory' and 'transcranial magnetic stimulation.' These terms were selected to be as broad as possible while aiming to exclude studies utilizing stimulation tech-niques (invasive and non-invasive) other than TMS. These terms were added to the Query Box using 'AND' Boolean operators; all possible fields (i.e. title, abstract) were searched. PubMed automatically included a search for similar terms for each of the terms of interest entered (i.e. 'memory' and 'memo-ries') and included the Medical Subject Headings (MeSH) terms associated with them. This search returned 104 results. A secondary search was conducted using Scopus after completing the PubMed search, using the same primary search terms using 'AND' Boolean operators and constraining fields to Article title, Abstract, and Keywords and limiting results to peer-reviewed empirical reports by limiting the 'Document Type' option to 'Article.'. This search returned 160 results. We also examined the references of publications identified as meeting the eligibility criteria for suggestions about other

relevant publications. Publications were selected from the search results when two reviewers (PFA and JLV) agreed that they met the two eligibility criteria. Additionally, only studies in human participants were selected.

## Outcomes extraction

Following review of the full texts, we extracted data from statistical tests aiming to quantify the effects of rTMS on episodic memory and non-memory outcomes. Extracted data included sample sizes, means, and standard deviations of the outcome variables for experimental and control measurements, as well as results from reported statistical tests. These data were used to calculate effect sizes, using the test statistics or group means and variances following established methods (*Turner and Bernard, 2006*). Cohen's d was used to measure the effect size of pairwise comparisons. Partial eta squared was used to measure the effect size of comparisons using analysis of variance or other general linear models. We did not include effects from tests that provided 'composite' measures spanning multiple cognitive domains (e.g. Montreal Cognitive Assessment, Mini-Mental State Examination, etc.) as these could not be readily classified as reflecting episodic-memory versus various non-memory functions. In tasks with multiple subscales (e.g. Wechsler Memory Scale, NIH Toolbox for Cognition, etc.), individual subscale scores were used and categorized based on the task format (e.g. recall versus recognition) and/or by the intended cognitive domain (e.g. episodic memory versus sustained attention). Effect sizes were computed such that improvement in performance (i.e. higher accuracy, faster response time in speeded non-memory tasks, or fewer errors) were positive and impairments (i.e. lower accuracy, slower responses, or more errors) were negative. Prior to input for meta-regression, all estimates of effect size were converted to Hedges' g to minimize potential impact of inflated effect sizes associated with small sample sizes (*Turner and Bernard, 2006*). Within and between estimation uncertainty around each Hedge's g were approximated using its corresponding study effective sample size.

## Meta-analysis of effect size

A mixed effects univariate meta-regression model estimated via restricted maximum likelihood was used to estimate the overall effect of rTMS on outcomes, pooling all effects reported in the final sample of studies collected at the outcome level and weighting them in terms of their within and between estimation uncertainty. We included a study random effect in the model to account both for inter and intra study variability in our final estimate, as a single study often reported various effect sizes for different outcomes. Heterogeneity in individual effect sizes was examined using a generalized Q-test to assess whether variability in observed effects exceeds what would be expected from random sampling error alone (*van Houwelingen et al., 2002*; *Viechtbauer, 2010*).

## Meta-analysis of effect modifiers

Factors that varied among studies were identified based on a priori hypotheses and *post hoc* consideration of study design details that could influence outcomes (Table in *Supplementary file 1*, *Figure 3*). Our categorization of factors was intended to include important design issues that could inform future interventional trials, such as the study population investigated, whether experiments used multi-session stimulation with interventional goals versus acute stimulation with basic-science goals, the delay at which outcomes were assessed, and the trial design and control conditions. The accompanying supplementary table describes these factors and the rationale for their inclusion (Table in *Supplementary file 1*).

We explored the association of effect size with these suspected modifiers by estimating a multivariate meta-regression model, including all study factors. As with the main effects model, we pooled all effects reported in the final sample of studies collected at the outcome level and weighted them in terms of their within and between study estimation uncertainty. Since some study characteristics were a priori expected to be highly correlated with one another (e.g. theta-burst stimulation is typically performed at lower intensity than high-frequency rTMS), we performed a previous-variable-selection step before assessing effect modification. Indeed, some factors were highly correlated, as expected (*Figure 3—figure supplement 1*). We, therefore, used a bootstrapped (1000 iterations) lasso model with a cross-validated penalization term to identify study characteristics that were jointly consistently redundant in terms of statistically explaining HITS effects (i.e. discarded by lasso with a probability

larger than 50% across all iterations), to be then excluded from the effect modification analysis. Study factors retained after the lasso variable selection were then simultaneously included in the multivariate mixed effects meta-regression, where their association with stimulation effects was estimated. Heterogeneity in individual effect sizes after accounting for effect modifiers was examined using a generalized Q-test, assessing whether significant variability in observed effects remained, even after accounting for the variation explained by included study characteristics (Q(df = 136)=307.07, $p<0.001$). Analysis for this step was conducted using the *glmnet* package in R. All meta-analytic estimates were computed using the *metafor* package in R (**Viechtbauer, 2010**).

## Outlier removal

To guard against the impact that unusually strong reported effects could have had in our findings, we repeated all meta-analyses, including the lasso variable selection step, after removing outliers identified using the studentized deleted residuals (**Viechtbauer and Cheung, 2010**). We identified and excluded seven effect outliers in the analysis of episodic memory outcomes, leaving 133 of 140 effects for that analysis, and seven effect outliers in the analysis of non-memory outcomes, leaving 106 of 113 effects for that analysis. Categorical variables were included in final models if at least one level had a coefficient with magnitude ≥|0.05| or their aggregate coefficients had magnitude ≥|0.01|. Variables with all coefficients <|0.01| were not included. We report findings with and without removal of outliers, which had little impact on overall results.

## Reporting bias assessment

Publication bias was assessed with a funnel plot of individual effect sizes against their standard errors (**Figure 3—figure supplement 2**; **Figure 3—figure supplement 3**). Special attention was paid to the asymmetry of the plot and the absence of effects in the lower-left corner of the funnel, which indicates the lack of published small-sample studies reporting negative effect sizes, further confirmed by an Egger's regression test ($p<0.001$), which is commonly carried out to detect evidence of publication bias (**Egger et al., 1997**). We tested for heterogeneity ($I^2$) to measure the inconsistency of effect sizes across studies within each cognitive domain. The $I^2$ statistic represents the percentage of variation in effect sizes beyond a random sampling error (**Higgins and Thompson, 2002**). Although the interpretation of heterogeneity can depend on multiple factors, an $I^2$ of 25 to 75% is typically considered moderate, whereas $I^2>75\%$ is considered a substantial heterogeneity (**Higgins and Thompson, 2002**). Further, we tested for the presence of outliers (described above), and removed them in sensitivity analyses to assess the robustness of our main findings to outliers. There was no conclusive evidence of publication bias, with $I^2$ of 0.53 for episodic memory tests and of 0.0 for non-memory tests (**Figure 3—figure supplement 2**; **Figure 3—figure supplement 3**) and with replication of major findings in the sensitivity analyses.

## Additional information

### Funding

No external funding was received for this work.

### Author contributions

Elena Badillo Goicoechea, Formal analysis, Methodology, Writing – original draft, Writing – review and editing; Phillip F Agres, Data curation, Formal analysis, Methodology, Writing – original draft, Writing – review and editing; Johanna MH Rau, Arantzazu San Agustin, Data curation, Writing – review and editing; Joel L Voss, Conceptualization, Data curation, Formal analysis, Writing – original draft, Project administration, Writing – review and editing

### Author ORCIDs

Joel L Voss ⬛ https://orcid.org/0009-0009-9028-1136

Reviewer #1 (Public review): https://doi.org/10.7554/eLife.108934.3.sa1
Reviewer #2 (Public review): https://doi.org/10.7554/eLife.108934.3.sa2

Reviewer #3 (Public review): https://doi.org/10.7554/eLife.108934.3.sa3
Author response https://doi.org/10.7554/eLife.108934.3.sa4

## Additional files

### Supplementary files
Supplementary file 1. Table of study factor descriptions.

MDAR checklist

### Data availability
A table of standardized effect sizes categorized by study and by the factor categories to which they were assigned for analysis, along with analysis code for reproducing findings are deposited in Dryad (datadryad.org), available for download at: https://doi.org/10.5061/dryad.vhhmgqp86.

The following dataset was generated:

| Author(s) | Year | Dataset title | Dataset URL | Database and Identifier |
|---|---|---|---|---|
| Badillo Goicoechea E, Agres P, Rau J, San Agustín A, Voss JL | 2026 | Data and code from: A meta-analysis suggests that TMS targeting the hippocampal network selectively improves episodic memory | https://doi.org/10.5061/dryad.vhhmgqp86 | Dryad Digital Repository, 10.5061/dryad.vhhmgqp86 |

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
