## [Editor Report · eLife Assessment]

This meta-analysis provides a **fundamental** synthesis of evidence demonstrating that transcranial magnetic stimulation targeting the hippocampal-cortical network reliably enhances episodic memory performance across diverse study designs. The evidence is **convincing**, with rigorous methodology and consistent effects observed despite modest sample sizes and some heterogeneity in stimulation approaches. The work highlights the specificity of memory improvements to hippocampal-dependent memories and identifies key methodological factors-such as individualized targeting-that influence efficacy. Overall, this study offers a timely and integrative framework that will inform both basic memory research and the design of future clinical trials for cognitive enhancement.

---

## [Referee Report · Reviewer #1 (Public review)]

Summary:

Goicoechea et al. conducted a timely and thorough meta-analysis on the potential for indirect hippocampal targeted transcranial magnetic stimulation (TMS) to improve episodic memory. The authors included additional factors of interest in their meta-analysis which can be used to inform the next generation of studies using this intervention. Their analysis revealed critical factors for consideration: TMS should be applied pre-encoding, individualized spatial targeting improves efficacy, and improvement of recollection was stronger than recognition.

Strengths:

As mentioned previously, the meta-analysis is timely and summarizes an emerging set of studies (over the past decade since Wang et al., Science 2014). Those outside of the field may not be aware of the robustness in improvements in episodic memory from hippocampal targeted TMS. The authors were quite thorough in including additional factors which are important for the interpretation of these findings. These factors also address the differences in approach across studies. The evidence that individualized spatial targeting improves TMS efficacy is consistent with recent advances in TMS for major depressive disorder. The specificity of the cognitive improvements to recollection of episodic memory and not for other cognitive domains is consistent with hippocampal targeting. The authors also plan to post the complete dataset on an open-source repository which enables additional analysis by other researchers.

Weaknesses:

The write-up is succinct and emphasizes the scientific decisions that underly key differences in the various experimental designs. While the manuscript is written for a scientific audience, the authors are likely aware that findings like this will be of broad appeal to the field of neurology where treatments for memory loss are desperately needed. For this reason, the authors could consider including a statement regarding an interpretation of this meta-analysis from a clinical standpoint. Statements such as 'safe and effective' imply a clinical indication and yet the manuscript does not engage with clinical trials terminology such as blinding, parallel arm versus crossover design, and trial phase. While the authors might prefer not to engage with this terminology, it can be confusing when studies delivering intervention-like five-days of consecutive TMS (e.g., Wang et al., 2014) are clustered with studies that delivered online rhythmic TMS which tests target engagement (e.g., Hermiller et al., 2020). While the 'sessions' variable somewhat addresses the basic-science versus intervention-like approach, adding an explicit statement regarding this in the discussion might help the reader to navigate the broad scope of approaches that are utilized in the meta-analysis.

Following revision: The authors have adequately addressed my concerns.

---

## [Referee Report · Reviewer #2 (Public review)]

Parietal lobe TMS, targeted to the episodic memory network via connections with the structures in the medial temporal lobe, improves episodic memory. This is one of very few robustly reproduced cognitive findings in noninvasive brain stimulation. The comprehensive review and detailed meta-analysis by Goicoechea et al. makes a convincing case for efficacy in healthy people and will be important for neuroscientists and clinical researchers in memory and dementia.

In 2014, Wang et al. showed that noninvasive stimulation of a parietal site, connected functionally to the hippocampus, increased resting state functional connectivity throughout a canonical network associated with episodic memory. It also caused a memory boost which was proportional to the connectivity increase within subjects. Their discovery that an imaging biomarker could (1) be used to target a functional network with critical nodes too deep to reach directly with TMS, (2) enable individualized, functionally confirmed, targeting, and (3) provide a scaling measure of target engagement, is one of the signal advances in noninvasive brain stimulation.

The meta-analytical methodology used by these authors is rigorous, and the central finding, viz. that high-frequency, network-targeted stimulation reproducibly improves event recall, is amply supported. The question of whether to stimulate before or after memory encoding is also answered. While there is a hint that individualized anatomical or functional MRI-based targeting may be superior to atlas or group average-based techniques, the finding did not survive correction for multiple comparisons. Additional studies will be needed to resolve this issue, optimize the stimulation delivery parameters, and further define the behavioral effect.

While the authors appropriately emphasize the associated network rather than the hippocampus itself, naming the target after a single node could suggest a primary role for the hippocampus in the observed outcomes, a conclusion not supported by the data reviewed here. Other nodes in the network are be equally important to aspects of episodic memory and could be useful targets for stimulation.

Despite encouraging results from small clinical samples, the question of efficacy in patients with static lesions and ongoing neurodegeneration remains open. The information gathered here, including the absence of reported adverse events, should spur Phase 2 clinical trials in patients with disorders of memory.

---

## [Referee Report · Reviewer #3 (Public review)]

Summary:

The manuscript by Goicoechea et al. assesses the influence of hippocampal-network targeted TMS to parietal cortex on episodic memory using a meta-analytic approach. This is an important contribution to the literature, as the number of studies using this approach to modulate memory/hippocampal function has clearly increased since the initial publication by Wang et al. 2014. This manuscript makes an important contribution to the literature. In general, the analysis is straightforward and the conclusions are well-supported by the results.

Strengths:

(1) A meta-analysis across published work is used to evaluate the influence of hippocampal-network-targeted TMS in parietal cortex on episodic memory. By pooling results across studies, the meta-analytic effects demonstrate an influence of TMS on memory across the diversity of many details in study design (specific tasks, stimuli, TMS protocols, study populations).

(2) Selectivity with regard to episodic memory vs. non-episodic memory tasks is evaluated directly in the meta-analysis.

(3) The investigation into supplemental factors as predictors of TMS's influence on memory was tested. This is helpful given the diversity of study designs in the literature. This analysis helps to shed light on which study designs, e.g., TMS protocols, etc., are most effective in memory modulation.

Weaknesses:

The authors thoroughly addressed and responded to the prior comments in the revision. The only minor weakness I see is acknowledged in terms of how null effects for particular design or TMS features should be interpreted (i.e., with caution given the regression approach used).

---

## [Author Response]

The following is the authors’ response to the original reviews.

**Reviewer #1 (Public review):**
(1) While the manuscript is written for a scientific audience, the authors are likely aware that findings like this will be of broad appeal to the field of neurology, where treatments for memory loss are desperately needed. For this reason, the authors could consider including a statement regarding an interpretation of this meta-analysis from a clinical standpoint. Statements such as 'safe and effective' imply a clinical indication, and yet the manuscript does not engage with clinical trials terminology such as blinding, parallel arm versus crossover design, and trial phase. While the authors might prefer not to engage with this terminology, it can be confusing when studies delivering intervention-like five days of consecutive TMS (e.g., Wang et al., 2014) are clustered with studies that delivered online rhythmic TMS, which tests target engagement (e.g., Hermiller et al., 2020). While the 'sessions' variable somewhat addresses the basic-science versus intervention-like approach, adding an explicit statement regarding this in the discussion might help the reader navigate the broad scope of approaches that are utilized in the meta-analysis.

We appreciate the suggestion to enhance interpretability of our report by broader audiences. First, to avoid confusion, we have eliminated “safe” and “effective” descriptors from the main summary of findings in the Abstract (pg. 1) and Discussion (pg. 6). Second, we now describe that reviewed studies included those categorized as traditional clinical trials, as well as non-clinical studies that generally follow clinical trial designs (i.e., multi-day intervention-like studies), in addition to more basic-oriented studies that are geared towards target engagement (Introduction, pg. 2). Third, we now clarify that the Design and Control factors (Figure 3) correspond to fairly standard distinctions in the clinical trials literature and were intended to capture major study design factors choices that are used in both clinical-trial and non-trial studies (Methods, pg. 9; Table S1). Finally, we now clarify that future clinical trials would be needed to evaluate HITS for any specific indication, and that our findings motivate such investigations but do not conclusively indicate efficacy for any given indication (Abstract, pg. 1; Discussion, pg. 7).

**Reviewer #1 (Recommendations for the authors):**
(1) The color scheme of Figure 1 was a bit confusing. All of the colors used for the flagged regions were incredibly similar. At first glance, it looks like the hippocampus was targeted directly due to the subtle color difference. Could the authors use colors that are more different? Similarly, zooming into the specific locations shows blue dots encompassed by teal. I am not sure what I am looking at here.

We have updated the figure for clarity.

(2) Given the broad appeal of the current study, I would encourage the authors to include a brief visual depiction of "HITS." This could help the more casual reader to understand the general approach.

We have included this in Figure 1A.

**Reviewer #2 (Public review):**
(1) While the introduction centers on the role of the hippocampus in episodic memory and posits hippocampal neuromodulation by TMS as causative, the true mechanism may be more complex. Clean hippocampal lesions in primates cause focal loss of spatial and place memory, and I am aware of no specific evidence that the hippocampus does more than this in humans. Moreover, there is evidence that lateral parietal TMS also reaches neighboring temporal lobe regions, which contribute to episodic memory. The hippocampus may, therefore, be a reliable deep seed for connectivity-based targeting of the episodic memory network, but might not be the true or only functional target.

We regret to have implied that we think the hippocampus is the true or only functional target. We agree with the reviewer that the hippocampus is “a reliable deep seed for connectivity-based targeting of the episodic memory network” and that the specific locus/loci of the HITS effects and mechanisms are not yet clear. We now emphasize that although hippocampus is used to define the targeted network, effects of TMS are likely distributed throughout the network, citing relevant studies that have shown that brain activity changes due to HITS are certainly not restricted to the hippocampus (Introduction, pg. 2).

(2) The meta-analysis combines studies with confirmation of targeting and target-network engagement from fMRI and studies without independent evidence of having stimulated the putative target (e.g., Koch et al). That seems like a more important methodological distinction than merely the use of any individual targeting method. In my experience, atlas-based estimates are at least as accurate as eyeballing cortical areas in individuals. Hence, entering individual functional targeting as a factor might reveal an effect on efficacy.

Our current definition of the “Targeting” factor appears to satisfy this concern. That is, we distinguish studies that used “individual functional targeting” (i.e., resting-state fMRI or DTI connectivity in each individual to select the target) from those that did not (i.e., atlas or other group-average approach). Notably, the Targeting factor modulation effect failed to survive correction for multiple comparisons. We think this satisfies the reviewer criticism, unless the reviewer is suggesting that we categorize studies based on whether they included evaluation of target engagement (e.g., tested for change in fMRI activity or connectivity of the network due to HITS) versus those that measured only behavioral outcomes. We did not include this distinction as a factor, as our analysis focuses on behavioral effects of HITS, and it is not clear what the neural effects would have been in studies in which they were not measured. Notably, we are providing the full raw dataset of effect sizes in a public repository with our final version of record, such that any other categorization schemes could be assessed by others.

(3) The funnel plot and Egger's regression for episodic memory outcomes suggested possible bias, and the average sample size of 23 is small, contributing to the likelihood of false positive results. It would be informative, therefore, to know how many or which studies had formal power estimates and what the predicted effect sizes were.

Regarding the average sample size of 23, we note that we used Hedges’ g for the effect size measure because it corrects for bias associated with small samples (pg. 10). Further, small sample sizes contribute to noisy estimates of true effects, allowing outliers to contribute to false positives and low power to contribute to false negatives, but without any reason to systematically yield bias towards false positives. Regarding potential publication bias, although we cannot rule this out based only on the statistics, we think that bias against publication of negative results is unlikely. First, HITS experiments are time consuming and expensive, and most in the field seem to be motivated to publish, whatever the outcome. Second, the notion of memory enhancement via brain stimulation is controversial, and groups have certainly been motivated, if not overly eager, to publish “failure to replicate” studies for HITS (e.g., the failure-to-replicate publication by Hendrikse et al. 2020, which was then re-analyzed by many of the original authors to arrive at different conclusions in Cash et al. 2022). Given these considerations, we think that it is very unlikely that publication bias had any major impact on our conclusions, but of course it cannot be conclusively excluded. Finally, we note that our finding of HITS selectivity for recollection enhancement is likely not affected by publication bias, as this selectivity versus other memory and non-memory outcomes was found only within published studies (i.e., it is very unlikely that publication bias would have led researchers to withhold publication of studies that found effects of HITS on recognition but not on recollection).

(4) In the Discussion, the authors might provide a comparison between the effect size for memory improvement found here with those reported for other brain-targeted interventions and behavioral strategies. It may also be worthwhile pointing out that HITS/memory is one of the very few, or perhaps the only, neuromodulatory effects on cognition that has been extensively reproduced and survived rigorous meta-analysis.

We now emphasize that this is, to our knowledge, the only neuromodulatory effect on cognition that is selective, has been extensively reproduced, and survived rigorous meta-analysis (Discussion, pg. 6). However, we wish to avoid the clinical overinterpretation of our findings that might result if we were to compare directly to effect size estimates for other current therapies, which have been evaluated for specific clinical indications. For example, antibody and pharmacological interventions for Alzheimer’s dementia typically have been associated with similar effect sizes to our estimate for HITS. However, those estimates derive from systematic review of randomized controlled trials measuring clinically relevant outcomes at relatively long delays, whereas the HITS studies we review include a mix of controlled and uncontrolled trials, vary in whether clinical outcomes were assessed, and mostly assessed outcomes at shorter delays. Thus, it could be misleading to directly compare the effect sizes. We instead continue to highlight that the HITS effects are promising and warrant rigorous testing for any given clinical indication.

(5) The section of the Discussion on specificity compares HITS to transcranial electrical stimulation without specifying an anatomical target or intended outcome. A better contrast might be the enormous variety of cognitive and emotional effects claimed for TMS of the dorsolateral prefrontal cortex.

We now also note that TMS of lateral frontal cortex has not been associated with similarly high specificity (Discussion, pg. 6). Note however that we cannot exclude anti-depressant or other psychological effects of HITS, as such outcomes were not consistently assessed in HITS studies and so were not included in our analyses.

(6) With reference to why other nodes in the episodic memory network have not been tested, current flow modeling shows TMS of the medial prefrontal cortex is unlikely to be achievable without stronger stimulation of the convexity under the coil, in addition to being uncomfortable. The lateral temporal lobe has been stimulated without undue discomfort.

We now additionally indicate that medial prefrontal stimulation may be ineffective given conventional TMS (Discussion, pg. 7). However, we are aware of no studies that have stimulated the portion of middle temporal gyrus that shows strong connectivity with hippocampus. We have tried this location, which positions the coil on or slightly above the ear and bordering on the temple area that is very sensitive to most. We were not able to minimize pain/discomfort for most subjects in pilot experiments, and so had to abandon it. Perhaps others have succeeded? If the reviewer has any specific references that could be included we would be happy to add them and update this section accordingly.

(7) Finally, a critical question hanging over the clinical applicability of HITS and other neuromodulation techniques is how well they will work on a damaged substrate. Functional and/or anatomical imaging might answer this question and help screen for likely responders. The authors' opinion on this would be informative.

We appreciate this point but don’t think there are enough data to assess the level of substrate damage needed to frustrate any stimulation benefits. The only thing we can say is that HITS was equally effective for mild to moderate Alzheimer’s dementia as it was for other non-neurodegenerative groups (nonsignificant effect of the Population factor, Figure 3B), suggesting that whatever degree of damage present in that group is insufficient to prevent the stimulation effects. We now highlight this point and raise the issue that, presumably, some level of damage would render HITS ineffective (Discussion, pg. 8).

**Reviewer #3 (Public review):**
(1) My only significant concern is how studies are categorized in the 'Timing' factor (when stimulation is applied). Currently, protocols in which TMS is administered across days are categorized as 'pre-encoding' in the Timing factor. This has the potential to be misleading and may lead to inaccurate conclusions. When TMS is administered across multiple days, followed by memory encoding and retrieval (often on a subsequent day), it is not possible to attribute the influence of TMS to a specific memory phase (i.e., encoding or retrieval) per se. Thus, labeling multi-day TMS studies as 'pre-encoding' may be misleading to readers, as it may imply that the influence of TMS is due to modulation of encoding mechanisms per se, which cannot be concluded. For example, multi-day TMS protocols could be labeled as 'pre-retrieval' and be similarly accurate. This approach also pools results from TMS protocols with temporal specificity (i.e., those applied immediately during encoding and not on board during memory testing) and without temporal specificity (i.e., the case of multi-day TMS) regarding TMS timing. Given the variety of paradigms employed in the literature, and to maximize the utility/accuracy of this analysis, one suggestion is to modify the categories within the Timing factor, e.g., using labels like 'Temporally-Specific' and 'Temporally Non-specific'. The 'Temporally-Specific' category could be subdivided based on the specific memory process affected: 'encoding', 'retrieval', or 'consolidation' (if possible). I think this would improve the accuracy of the approach and help to reach more meaningful conclusions, given the variety of protocols employed in the literature.

We agree in principle with this criticism and think that the most straightforward way to address it is to relabel the “Pre-Encoding” category as “Pre-Task”. The issue with labeling/considering single-session stimulation delivered immediately before encoding as “Pre-encoding” is that this makes the assumption that this stimulation doesn’t also affect retrieval (i.e., is temporally specific). We do not have certainty about the timecourse of how a single session of stimulation affects brain activity. We think the “Pre-Task” label and interpretation is the best way to address this, to avoid suggesting that we are confident about the timecourse/selectivity of stimulation effects. Notably, the “Sessions” factor directly compares among designs that delivered stimulation in a single session versus in multiple consecutive sessions, and was a nonsignificant modulator. Thus, our analyses already compare studies that are relatively temporally specific versus those that, likely, are less so. In addition to relabeling, we have also added clear caveats to address the interpretive constraint imposed by the unknown timecourse of stimulation effects (Discussion, pg. 6-7) and revised the Abstract to reflect this change.

(2) As the scope of the meta-analysis is limited to TMS applied to parietal or superior occipital cortex, it is important to highlight this in the Introduction/Abstract. The 'HITS' terminology suggests a general approach that would not necessarily be restricted to parietal/nearby cortical sites.

This was previously highlighted only in the Methods and Discussion (with a Discussion paragraph dedicated to the issue of target selection; see also Comment 6 from Reviewer 2). We now also note this in the Introduction (pg. 2) and Abstract.

Minor:(1) To reduce the number of study factors tested, data reduction was performed via Lasso regression to remove factors that were not unique predictors of the influence of TMS on memory. This approach is reasonable; however, one limitation is that factors strongly correlated with others (and predict less unique variance) will be dropped. This may result in a misrepresentation, i.e., if readers interpret factors left out of this analysis as not being strongly related to the influence of TMS on memory. I do see and appreciate the paragraph in the Discussion which appropriately addresses this issue. However, it may be worth also considering an alternative analysis approach, if the authors have not already done so, which explicitly captures the correlation structure in the data (i.e., shown in Figure S2) using a tool like PCA or an appropriate factor analysis. Then, this shared covariance amongst factors can be tested as predictors of the influence of TMS - e.g., by testing whether component scores for dominant PCs are indeed predictive of the influence of TMS. This complementary approach would capture rather than obfuscate the extent to which different factors are correlated and assess their joint (rather than independent) influence on memory, potentially resulting in more descriptive conclusions. For example, TMS intensity and protocol may jointly influence memory.

We argue that feature selection via Lasso regression is a better approach for our research question than PCA, factor analysis, or other latent variable methods. The main reason is that PCA would sacrifice the interpretability of our findings with respect to the design of future experiments using or testing HITS. That is, because PCA creates composite components that are linear combinations of multiple variables, we would lose the ability to provide clear, actionable guidance to researchers about which specific study design choices (e.g., stimulation intensity, protocol type, timing) influence memory outcomes. Given that a major goal of our meta-analysis is to inform future experimental design, we believe that it is essential to maintain interpretability of the individual factors that must be decided when designing a study. Regarding factor analysis, this approach would require making a *priori* theoretical decisions about how to group individual moderators, which could introduce subjective bias into the analysis and would introduce other complications such as a need for validation of the resulting factor scores. We believe that the exploratory nature of our investigation, examining which among many possible study design factors substantially determine TMS efficacy, is better suited to a data-driven selection approach like Lasso. While the reviewer correctly notes that Lasso may drop factors that are correlated with stronger predictors, this feature can be considered advantageous in terms of identifying factors for inclusion in future study designs. That is, this can help identify the most parsimonious set of independent predictors, such that researchers can focus on the study design elements that matter most when controlling for other factors. Notably, we provide the table of factor relationships (Figure S2) so that interested readers can inspect how dropped factors were related to those that were retained.

It is also important to note that we have provided the full dataset with our resubmission, which has been deposited in Dryad with a link in the Data Availability section (pg. 15). Thus, others are free to explore alternative analytical approaches should they wish to examine the data from different perspectives or to answer different questions.

(2) Given the specific focus on TMS applied to parietal cortex to modulate hippocampal and related network function, it would be fruitful if the authors could consider adding discussion/speculation regarding whether this approach may be effectively broadened using other stimulation methods (e.g., tACS, tDCS), how it may compare to other non-invasive brain stimulation methods with depth penetration to target hippocampal function directly (transcranial temporal interference, or transcranial focused ultrasound), and/or how or whether other stimulation sites may or may not be effective.

We briefly discuss a meta-analysis of tACS studies which reported nonspecific effects, including for parietal targets overlapping those used for HITS (Discussion, pg 6). We briefly speculate about how tES effects remain mechanistically uncertain. We are afraid that further speculation about other stimulation modalities and targets would be beyond the scope of this focused meta-analysis, given especially the few datapoints for newer approaches such as TI or tFUS.

(3) Studies were only included in the meta-analysis if they contained objective episodic memory tests. How were studies handled that included both objective and subjective memory, or other non-episodic memory measures? For example, Yazar et al. 2014 showed no influence of TMS on objective recall, but an impairment in subjective confidence. I assume confidence was not included in the meta-analysis. Similarly, Webler et al. 2024 report results from both the mnemonic similarity task (presumably included) and a fear conditioning paradigm (presumably excluded). Please clarify in the methods how these distinctions were handled.

Studies were included in our meta-analysis if they included at least one objectively scorable test of episodic memory. We only included objectively scorable test performance in our analysis, excluding scores from any other subjective measures if they were also reported. This is now clarified in Methods (pg. 9).

(4) The analysis comparing memory to non-memory measures is important, showing the specificity of stimulation. Did the authors consider further categorizing the non-memory tasks into distinct domains (i.e., language, working memory, etc.)? If possible, this could provide a finer detail regarding the selectivity of influences on memory vs. other aspects of cognition. It is likely that other aspects of cognition dependent on hippocampal function may be modulated as well, i.e., tasks with high relational/associative processing demands.

This is an interesting idea, but it is beyond our expertise to categorize these other tasks based on the nature of processing demands that they capture. Note that the task names are provided in the data table that we are making available online with our submission of record (via Dryad), such that other groups could address this question if interested.

(5) In the analysis of the Intensity factor, how were studies using Active (rather than resting) MT categorized? Only resting MT is mentioned in Table S1. This is important as the original theta-burst TMS protocol from Huang et al. 2005 determines intensity based on Active Motor Threshold.

MT was resting/passive in all reviewed studies except for one (Tambini et al. 2018), which used 80% of active MT. We categorized this as <100% MT for the Intensity factor, as it was <100% of MT as defined in that study. Although one could make the argument that 80% AMT might instead correspond to 100+% RMT, this change would have very little influence on our results or conclusions. We now clarify this in Table S1.

(6) Is there a reason why the study by Koen et al. 2018 (Cognitive Neuroscience) was not included? TMS was performed during encoding to the left AG, and objective memory was assessed, so it would seemingly meet the inclusion criterion.

The failure to include Koen et al. 2018 was our error. Koen et al. 2018 is the only study that used “online” stimulation, delivered during the trials when memoranda were displayed for encoding in the task. In contrast, all other reviewed studies delivered “offline” stimulation either before the memoranda was presented (“Pre-Task”) or after the encoding period but before retrieval (“Post-Encoding”). Therefore, categorization for the “Timing” factor would be problematic for its inclusion in the main analysis. We therefore now include Koen et al. 2018 in the “Supplementary Results” section as well as the corresponding main Results section on “Similar outcomes in studies that were excluded from meta-analysis”. We also note in the relevant discussion that “online” stimulation, as done in Koen et al. 2018, is typically considered disruptive (e.g., Beynel et al. 2019 Neuroscience & Biobehavioral Reviews; Yeh & Rose 2019 Frontiers in Psychology), which should be taken into account when considering the findings of Koen et al. 2018 relative to other reviewed studies that used “offline” designs.

(7) It would be helpful to briefly differentiate the current meta-analysis from that performed by Yeh & Rose (How can transcranial magnetic stimulation be used to modulate episodic memory?: A systematic review and meta-analysis, 2019, Frontiers in Psychology) (other than being more current).

Beyond being more current and therefore including many more studies in which stimulation targets were based on hippocampal connectivity (which tend to have been published more recently), the differences with Yeh & Rose 2019 are subtle. Our review focuses on assessment of network targeting and whether effects were specific to episodic memory versus other tasks, which differs somewhat from the focus of Yeh & Rose 2019. The main difference in conclusions likely derives from there being more network-focused memory TMS experiments now than were available for Yeh & Rose’s review. We also differentiate episodic memory into recollection versus other components to test specificity and analyze modulation by many study design factors relevant to HITS studies that were not emphasized in Yeh & Rose’s review. Note that we now cite Yeh & Rose for those interested in potential differences.

(8) For transparency and to facilitate further understanding of the literature and potential data re-use, it would be great if the authors consider sharing a supplementary table or file that describes how individual studies/memory measures were categorized under the factors listed in Table S1.

As promised in our original submission, we are providing the full data table, including how individual studies and memory measures were categorized, as an open dataset in Dryad. The Dryad dataset is cited in “Data availability” (pg. 15).

**Reviewer #3 (Recommendations for the authors):**
Please explicitly state in the Methods (Meta-analysis of effect modifiers section) that the criteria used for categorizing each measure into a factor (e.g., probing Recollection, Recognition, etc.) are fully described in Table S1; this will help readers to find these details (it took me a while!).

This is now emphasized (pg. 10).